# Brown Fat and Nutrition: Implications for Nutritional Interventions

**DOI:** 10.3390/nu15184072

**Published:** 2023-09-20

**Authors:** Lloyd Noriega, Cheng-Ying Yang, Chih-Hao Wang

**Affiliations:** 1Graduate Institute of Biomedical Sciences, College of Medicine, China Medical University, Taichung 406040, Taiwan; 2Graduate Institute of Cell Biology, College of Life Sciences, China Medical University, Taichung 406040, Taiwan

**Keywords:** metabolic diseases, mitochondria dysfunction, obesity, diabetes, fatty liver disease, white adipose tissue, brown adipose tissue, beige adipose tissue, UCP1, nutritional interventions

## Abstract

Brown and beige adipocytes are renowned for their unique ability to generate heat through a mechanism known as thermogenesis. This process can be induced by exposure to cold, hormonal signals, drugs, and dietary factors. The activation of these thermogenic adipocytes holds promise for improving glucose metabolism, reducing fat accumulation, and enhancing insulin sensitivity. However, the translation of preclinical findings into effective clinical therapies poses challenges, warranting further research to identify the molecular mechanisms underlying the differentiation and function of brown and beige adipocytes. Consequently, research has focused on the development of drugs, such as mirabegron, ephedrine, and thyroid hormone, that mimic the effects of cold exposure to activate brown fat activity. Additionally, nutritional interventions have been explored as an alternative approach to minimize potential side effects. Brown fat and beige fat have emerged as promising targets for addressing nutritional imbalances, with the potential to develop strategies for mitigating the impact of metabolic diseases. Understanding the influence of nutritional factors on brown fat activity can facilitate the development of strategies to promote its activation and mitigate metabolic disorders.

## 1. Introduction

Metabolic diseases such as obesity, type 2 diabetes, nonalcoholic fatty liver disease, osteoporosis, gout, hyperthyroidism, and hypothyroidism are becoming major global health issues, affecting not only the health of billions of adults worldwide but also the economy [1]. In 2019, the medical care for obesity in the US alone was nearly USD 173 billion, and diabetes has become the 9th leading cause of death worldwide [1]. Given the impact of these metabolic diseases, understanding the role of nutrition in these diseases, specifically addressing nutritional imbalances, becomes crucial.

The nutritional imbalance has detrimental effects on adipose tissue, leading to disrupted function and an increased risk of metabolic disorders [2,3]. It promotes excessive fat storage, chronic inflammation, altered lipid metabolism, dysregulated adipokine secretion, and compromised brown fat activity. These changes contribute to the development of metabolic disorders, emphasizing the importance of a balanced diet for maintaining healthy adipose tissue and overall metabolic well-being [4]. Existing pharmaceutical interventions primarily focus on reducing energy intake by suppressing appetite or impeding the absorption of dietary fats. Several drugs using this strategy, such as orlistat and naltrexone/bupropion, are approved by US FDA. However, the lack of effective drugs for increasing energy expenditure still requires effort.

A groundbreaking discovery of a novel type of adipose tissue, known as brown adipose tissue (BAT), has emerged as a promising avenue to address nutritional imbalances within the human body [4]. This remarkable finding unveils new possibilities for tackling nutritional challenges and presents exciting prospects in the field of metabolic research [5,6]. As individuals progress through adulthood, there is a notable decline in the quantity of BAT, commonly referred to as brown fat [7]. This decrease in BAT content is observed to be negatively correlated with body mass index (BMI), a measure used to assess an individual’s body composition and overall weight status [8]. The relationship between age and BAT reduction suggests a potential role of age-related physiological changes in the decline of BAT and its implications for body weight regulation. Understanding the factors influencing BAT decline and its connection with BMI may contribute to our knowledge of metabolic changes occurring during aging and the potential impact on overall health and weight management strategies [6,9].

Brown fat has therapeutic potential, which can reverse certain phenotypes in diabetic patients. Cold exposure has been identified as the most natural and efficient method to activate and mobilize BAT [10]. During cold stress, thermogenesis is classically stimulated by norepinephrine released from the sympathetic nervous system, which activates β3-adrenergic receptors in brown adipocytes [7]. However, cold therapy is not always convenient [10]. Achieving well-controlled conditions for human cold exposure, especially considering the presence of clothing and heating systems, can be challenging and uncomfortable. Additionally, prolonged cold exposure or a drug such as mirabegron mimics the effect of cold exposure, has been associated with an elevation in heart rate and blood pressure [11].

Another viable option to activate brown fat activity is a nutritional diet, which is safer and more practical [12]. This approach aims to minimize the side effects of other alternative treatments, making nutritional supplements a more favorable option with fewer adverse effects. Exploring the implications of nutritional interventions on brown fat activation, energy expenditure, and metabolic regulation holds promise for developing strategies to mitigate the impact of metabolic diseases. This review summarizes different strategies to promote thermogenic fat in the human body, hoping it will help inspire future research.

## 2. Overview of Different Adipose Tissues

Adipose tissues play a major role in the human body processes such as energy storage, immune release, and heat generation. Adipose tissue can be categorized into various types and phenotypes. In mammals, three main types of adipose tissue are simplified classifications: white adipose tissue (WAT), BAT, and beige adipose tissue (beige fat) [13]. They are different in morphology, distribution, mitochondrial content, and functions (Figure 1). WAT can be subdivided into four distinct adipose tissue types, which primarily include subcutaneous WAT, visceral WAT, epicardial WAT, and perivascular WAT. These different types represent various functional and anatomical states of white adipocytes [14]. In summary, the distinctions of these six adipose tissues aid in comprehending the diverse roles of adipose tissue in metabolic processes and overall well-being.

The adipocyte is the principal cell of adipose tissue, which contains lipid droplets. The adipocyte stores energy as a triglyceride. Glycogen is a stored form of glucose primarily found in the liver and muscle cells. Thermogenic adipocytes also store energy in the form of glycogen, particularly after birth. When energy demands are high, such as during physical activity, glycogen can be broken down through a process called glycogenolysis to release glucose for immediate energy use. However, when glycogen stores are sufficient and energy demands are low, excess glucose can be converted into triglycerides, which are a form of stored energy in adipocytes. This process is known as lipogenesis [15,16].

WAT is a type of adipose tissue that stores energy in our body. The origin has been extensively studied and is believed to be from a population of mesenchymal progenitor cells that can differentiate into adipocytes [14]. These progenitors, in turn, differentiate into mature white adipocytes in response to various factors, including insulin, glucocorticoids, and sex steroids [13,15,17]. In response to appropriate stimuli such as cold exposure and β3-adrenergic agonists, WAT undergoes browning, wherein it acts similarly to BAT.

BAT was seen in small mammals during their hibernation and in small infants, which utilize BAT for non-shivering thermogenesis [18]. Like WAT, BAT originates from the embryonic mesoderm. BAT develops from Myf5 precursor cells. The study used genetic lineage tracing to show that BAT originates from a specific population of progenitor cells that express the transcription factor Myf5, which is thought only from myogenic cells. The research article showed that PRDM16 controls the cell fate switch between skeletal myoblasts and brown fat cells [19]. BAT expresses uncoupling protein 1 (UCP1) located on the inner membrane of mitochondria to exert the thermogenesis process [20]. Compared to WAT and beige fat, it contains a significant number of mitochondria [17].

Beige fat, also known as brite (brown-in-white) or BAT-like fat, is a relatively new discovery and functions the same as BAT [21]. The origin of beige fat is not yet fully understood. However, research has shown that beige adipocytes can arise from several sources, including white adipocytes and tissue-resident adipocyte progenitors. Beige fat can be activated from WAT in response to various stimuli, such as cold exposure, exercise, and certain hormones [13,17,18,22].

Thermogenic adipose tissues such as BAT and beige fat contain abundant mitochondria filled with UCP1 and dissipate energy generated from circulating fatty acids/glucose in the form of heat by uncoupling the proton gradient on the mitochondria inner membrane, a process known as thermogenesis [23,24]. Knockout of UCP1 in mice resulted in cold sensitivity [25], which further proves that UCP1 oversees non-shivering thermogenesis [20,26,27].

UCP1-mediated thermogenesis is essential for classical non-shivering thermogenesis and cold acclimation-induced thermogenesis. However, recent studies have shown that UCP1 inactivation has occurred in multiple placental mammal clades, leading to a loss of thermogenic capacity. In animals living under constant low temperatures, the UCP1 expressing thermogenic fat is lost, as heat insulation by WAT is a better adaptation strategy to cold than non-shivering thermogenesis or skeletal muscle heat generation [28,29,30]. These genetic changes could be the result of evolutionary adaptations to specific environmental conditions or changes in dietary patterns. The same trend is also happening to the human population living in cold temperatures [31].

In addition to UCP1, recent findings have demonstrated that UCP1-independent pathways, such as the futile creatine cycle [32] or calcium recycling [26], play indispensable roles in thermogenesis. Additionally, they play important roles in fatty acid metabolism, multilocular lipid droplet formation in thermogenesis, and glucose uptake. BAT and beige fat are highly vascularized and metabolically active. Upon stimulation, they can enhance energy expenditure and increase glucose and fatty acid uptake [33]. Furthermore, brown and beige adipocytes act as metabolic sinks for nutrients such as glucose, fatty acids, and amino acids [34].

## 3. BAT Acts as a Therapeutic Target for Combating Metabolic Disorders

### 3.1. Distribution of BAT in the Human Body

It has been previously reported that approximately 45–135 g of BAT are present in adult humans, which represents about 0.6–0.8% of the total body mass. However, during cold exposure, this amount can increase to 40–50% [35]. BAT is distributed across different regions of the human body. In healthy adult humans, the perirenal and perivertebral fats represent active brown fat depots in the visceral region [36]. However, in both rodents and humans, BAT is found scattered within WAT [37]. In Madelung disease, lipomatous fat deposits may stem from dysfunctional brown adipose tissue [38]. Recent research has shown the distribution and persistence of BAT in 18 different body areas in English infants, children, younger adults, and elderly adults [39,40]. BAT activity is stimulated by exposure to cold, diminishes with age, and is inversely related to BMI [10,41]. In humans, BAT is abundant during infancy but diminishes during adulthood. An adult human is primarily composed of beige/brite cells in contrast to brown fat [42,43]. A significant quantity of beige adipocytes is observed in adult humans, particularly following exposure to cold temperatures [42,44]. Following the recent discovery of novel transcription factors controlling brown adipocyte differentiation and the existence of active brown fat in adult humans, the field of research on brown fat has attracted significant attention and is expanding quickly [45].

Furthermore, it is important to note that thermogenic fat depots can develop and be functional without cold exposure. Newborns, for example, have extensive thermogenic fat depots that do not require cold stress to be functional, and the breastmilk contains alkylglycerols, which can maintain the beige adipocytes, which progressively replace white adipocytes in late infancy and in adolescence [46,47], that is also observed in mice [48]. This indicates that cold exposure is not necessary for the development and functionality of thermogenic fat in infants [22]. The subcutaneous WAT in newborns is capable of energy expenditure and thermogenesis [49,50,51]. It contains gene products related to brown adipocytes, mitochondrial biogenesis, and the differentiation of thermogenic fat, which are expressed in the subcutaneous adipose tissue of newborns [52]. The presence of these gene products implies that newborns might possess functional brown and beige adipocytes, playing a role in generating heat and managing energy metabolism [53]. According to classical concepts, BAT develops from the 5th gestational week onwards, peaking around birth and declining over the next 9 months of extrauterine life [47,53].

### 3.2. Correlation between BAT, Nutritional Imbalance and Metabolic Disorders

Thermogenic fats have garnered significant research interest in recent years due to their potential as therapeutic targets for metabolic disorders such as fatty liver, obesity, and type 2 diabetes. An imbalance in nutrition can reduce the levels of thermogenic fats, which can contribute to the development of metabolic disorders. Studies have shown that obesity is associated with a deficiency in BAT and beige fat, and restoring those fats may be beneficial for metabolic health [54]. Inflammation associated with obesity and metabolic disorders can impair the thermogenic and metabolite drainage functions of BAT and beige fat. This impairment can contribute to the development of metabolic syndrome and cardiovascular alterations [55]. Another study has shown that the miR-199a/214 cluster acts as a significant suppressor of the development and thermogenic activity of BAT and beige fat. In obesity, the miR-199a/214 cluster is upregulated, inhibiting the browning of adipose tissue through the PRDM16-PGC-1α transcriptional network, which is important for the brown fat lineage [19,56].

The presence of BAT is associated with improved cardiometabolic health, including healthier body fat distribution, improved glucose and lipid metabolism, and decreased liver fat accumulation [45,57]. The positive impacts of BAT were particularly noticeable in individuals who were overweight or obese, suggesting that BAT could potentially help alleviate the harmful consequences of obesity. Collectively, these discoveries underscore the potential role of BAT in enhancing cardiovascular and metabolic well-being [58].

### 3.3. Strategies for BAT Activation

The activation of thermogenic adipocytes has demonstrated its ability to enhance glucose metabolism, reduce fat, and improve insulin sensitivity in preclinical models. The most effective method for activating BAT is through exposure to cold temperatures, which also triggers the browning of WAT into beige fat. However, this approach is not practical or convenient, particularly in countries with year-round hot weather where daylight exposure can deactivate thermogenic adipocytes [59]. An increasing number of studies are exploring the use of drugs that activate molecular pathways involved in thermogenesis, such as β3-andrenergic agonists that directly activate BAT/beige fat for inducing thermogenesis. Mirabegron, a β3-adrenergic receptor agonist approved by the FDA for treating overactive bladder, has been found to have effects on BAT and energy expenditure. Acute treatment with mirabegron increased BAT glucose uptake and energy expenditure in healthy humans [11]. However, a single 200 mg dosage was associated with elevated blood pressure, leading to a limitation on the clinical use of dosages higher than 50 mg per day. Lower doses of mirabegron, such as 100 mg for 4 weeks or 50 mg for 12 weeks, have been shown to activate thermogenic adipose tissue; reduce inflammation in adipose tissues and skeletal muscle; and improve glucose tolerance, insulin sensitivity, and pancreatic β-cell function. These lower doses have fewer adverse effects on cardiovascular function [60,61]. Additionally, transplantation is a newly proposed method, and favorable results have been observed in preclinical studies on small mammals [62].

Another approach to activate BAT/beige fat is through dietary modifications. By strategically altering our eating habits, such as intermittent fasting, caloric restriction, or adopting a ketogenic diet, our bodies can produce and activate BAT/beige fat to promote weight loss and harness its therapeutic benefits [63,64,65,66]. Therefore, the search for a more promising and effective means of targeting these cells is ongoing, although a comprehensive understanding of their development and function at the molecular level is still required. The molecular mechanisms underlying the differentiation of brown and beige adipocytes from precursor cells, as well as the browning phenomenon of WAT into beige fat, remain incompletely understood, necessitating further research to elucidate the transcriptional and epigenetic factors regulating these processes.

## 4. Nutritional Factors Regulate BAT Activity

Nutrition plays a significant role in the regulation of BAT. Several studies have shown that dietary factors can promote the development and activation of BAT and beige fat [23,67]. Altered BAT function contributes to the pathophysiology of obesity and glucose metabolism dysfunctions. The effect of high-fat diets on thermogenesis in brown adipose tissue may relate to the suppressive effects of dietary lipids on fatty acid synthesis in adipose tissue [68]. Furthermore, the investigation into the roles of macronutrients, micronutrients, and natural products in BAT regulation is still in its early stages (Table 1). Many challenges remain, including identifying safe and effective methods for activating thermogenic adipocytes in humans, as well as understanding the potential side effects and long-term safety of such interventions. 

### 4.1. Macronutrients and Their Impact on BAT

#### 4.1.1. Proteins

A high-protein diet is already a well-known strategy to lose weight, develop muscle mass, and decrease fat volume in the body [88]. This increased energy expenditure is attributed to the higher metabolic demands of protein metabolism and can contribute to overall energy balance and metabolic rate. Published literature proves that protein intake may influence the browning of WAT, including increased thermogenic activity and metabolic health [69,70,71].

#### 4.1.2. Carbohydrates

Carbohydrates are associated with BAT in various ways. In a study using a mouse model, consuming a high-carbohydrate meal produces thermogenesis compared to a high-fat meal, and the magnitude of this thermic effect corresponds to the difference in BAT thermogenesis [89]. A study has found that changes in the mitochondrial proton-conductance pathway in BAT may be responsible for the effects of carbohydrates [90]. However, decreased triglyceride content and increased carbohydrate utilization in brown adipocytes have been observed with the deletion of inducible nitric-oxide synthase in leptin-deficient mice [91]. Both classical brown and beige adipocytes break down stored lipids and carbohydrates to generate heat through a UCP1 [92].

#### 4.1.3. Omega-3 Fatty Acids

Omega-3 polyunsaturated fatty acids (PUFA) have been shown to have a positive effect on BAT activation and thermogenic activation. Studies have shown that omega-3 PUFA can induce brown and beige adipocyte differentiation and thermogenic function, and these effects require G-protein coupled receptor 120 (GPR120) [93]. Furthermore, endogenous omega-3 PUFA has been found to preserve the morphology and function of brown fat impaired by high-fat diet feeding in mice [94]. Clinical trials have investigated the effects of omega-3 fatty acids on body composition and lipid profile in elite athletes and found that increased omega-3 supplementation stimulates fat loss in individuals who experience obesity, diabetes, and metabolic syndrome [95,96]. Nevertheless, further investigation is necessary to identify the mechanisms underlying these effects and to establish the ideal dosage and duration of omega-3 supplementation to enhance BAT function and decrease the chances of obesity and associated metabolic disorders.

#### 4.1.4. Omega-6 Fatty Acids

Omega-6 fatty acids have been shown to have negative effects on brown fat. Experimental studies have suggested that omega-6 PUFA leads to white fat deposits and obesity, while omega-3 fatty acids increase thermogenesis through the expression of UCP1 and increase fatty acid oxidation [73,79,97]. A study found that a balanced diet of omega-6 and omega-3 fatty acids increased the mass of axillary brown adipose tissue in Arctic ground squirrels [98]. Reducing the intake of omega-6 fatty acids in the human diet could potentially enhance the activity of brown adipose tissue. This may be attributed to the competition between enzymes involved in processing omega-3 and omega-6 fatty acids, although additional research is required to explore this further [72]. Therefore, it is crucial to maintain a balanced ratio of omega-6 to omega-3 fatty acids to promote the browning of adipose tissue and prevent obesity.

### 4.2. Micronutrients and Their Impact on BAT

#### 4.2.1. Vitamin D

Vitamin D is well proven to be involved in various processes such as energy metabolism, antioxidant defense, adipocyte differentiation, and apoptosis [99]. Several studies have suggested that vitamin D could promote thermogenesis by promoting brown fat activity [99,100]. Obese individuals might be at a higher risk of vitamin D deficiency. One reason for this is that vitamin D is a fat-soluble vitamin, and it can get sequestered in body fat, potentially leading to lower circulating levels in individuals with higher body fat percentages [101,102]. A study also demonstrated that vitamin D activates mitochondrial biogenesis by inhibiting the interferon response to cytosolic mitochondrial RNA in the beige adipose tissue of newborns [103]. Vitamin D insufficiency also downregulates the expression of UCP1 in WAT and inhibits AMPK/SIRT1 activity, leading to obesity progression in rats [74]. In addition, fatty acid elongase 3 (ELOVL3), which is important for fatty acid metabolism in BAT, is directly regulated by vitamin D nuclear hormone receptor (VDR), suggesting that vitamin D and VDR play a special role in brown fat activity [104]. The expression of VDR in adipocytes has been shown to be involved in adipose tissue energy metabolism. In genetically modified mouse models, both VDR knockout and CYP27B1, an enzyme converting vitamin D to its active form, knockout models have shown resistance to diet-induced weight gain, while the specific overexpression of human VDR in adipose tissue leads to increased adipose tissue mass [105]. The synthesis of vitamin D is reliant on exposure to sunlight, and there is a noteworthy association between an individual’s BMI as an adult and the season during which they were born [106,107]. Another research has also shown that vitamin D, when it binds to VDRs in muscle cells, could potentially influence the expression of various genes, including UCP3. This means that the presence of sufficient vitamin D and active VDRs might impact the production of UCP3 in muscle cells [108]. However, further research is still needed to pinpoint the exact mechanism and interplay in the molecular pathway.

#### 4.2.2. Succinate

Research has indicated a potential relationship between succinate and BAT activity. Succinate is an intermediate molecule involved in cellular energy metabolism and the tricarboxylic acid cycle, also known as the Krebs cycle or citric acid cycle [109]. Succinate accumulation occurs independently of adrenergic signaling. Studies have suggested that succinate can modulate brown adipocyte function and thermogenesis. Brown adipocytes rapidly metabolize succinate by succinate dehydrogenase, which is essential for triggering thermogenesis. The oxidation of succinate generates reactive oxygen species, initiating thermogenic respiration. In contrast, inhibiting succinate dehydrogenase suppresses thermogenesis. Additionally, circulating succinate enhances UCP1-dependent thermogenesis in BAT in vivo. This stimulation of thermogenesis provides significant protection against diet-induced obesity and improves glucose tolerance [75].

#### 4.2.3. Inosine

Inosine has been published as a crucial molecule involved in energy expenditure in brown and beige adipocytes. When dying brown adipocytes release nucleotides and nucleosides, inosine is found in the highest volume. The study revealed that inosine activates cAMP/protein kinase A (PKA) signaling and increases thermogenesis in brown adipocytes. Furthermore, the researchers identified equilibrative nucleoside transporter (ENT1) as a protein regulator of inosine levels in adipocytes. These findings suggest that targeting ENT1 or increasing inosine could present an alternative approach for brown fat therapy [57,76].

#### 4.2.4. Arginine

Arginine is an amino acid that is involved in different roles in our bodies, such as protein synthesis, energy metabolism, and nitric oxide synthesis [110,111]. In obese mice, supplementation of L-arginine resulted in reduced white fat gain, increased skeletal muscle mass, decreased blood glucose and triglycerides, and improved insulin sensitivity [77]. The growth and development of BAT in rats are promoted by l-arginine through mechanisms involving nitric oxide signaling [111]. However, in Europid and South Asian overweight and prediabetic men, l-arginine supplementation has no impact on basal metabolic rate, BAT activity, or skeletal muscle mitochondrial respiration [112]. The connection between arginine and brown fat is still an area of scientific investigation, and more studies are required to establish definitive conclusions.

#### 4.2.5. 12,13-diHOME

The studies employed liquid chromatography-tandem mass spectrometry to analyze lipid profiles in response to cold exposure [78] and exercise [113]. They discovered that cold exposure and exercise led to an elevation in a specific thermogenic lipokine known as 12,13-diHOME. 12,13-diHOME promotes fatty acid uptake and utilization in BAT and skeletal muscle for the thermogenesis process. Furthermore, the researchers found a negative correlation between 12,13-diHOME levels and circulating triglycerides and leptin. The negative correlation implies that 12,13-diHOME may be associated with the regulation of lipolysis, energy expenditure, and body weight gain. The presence of 12,13-diHOME in human milk has been found to be associated with infant adiposity and body composition. A study investigated the association between 12,13-diHOME in human milk and infant adiposity. The researchers found that higher milk 12,13-diHOME was associated with increased weight-for-length at birth, lower infant fat mass at 1 month, and reduced gain in BMI from 0 to 6 months. These findings suggest that 12,13-diHOME in human milk may play a role in regulating infant adiposity [114].

#### 4.2.6. 12-HEPE

Omega-3 fatty acids, found in fish oil and other dietary sources, have been associated with increased BAT activity and thermogenesis [115]. They have shown the potential to induce the browning of WAT, enhancing its thermogenic capacity. One of the metabolites derived from 12-lipoxygenase (12-LOX) and omega-3 fatty acids, particularly eicosapentaenoic acid (EPA), is known as 12-hydroxyeicosapentaenoic acid (12-HEPE). It functions as a signaling molecule within the body, acting locally (paracrine) or systemically (endocrine) to enhance the uptake of glucose by BAT and skeletal muscle [79]. 12-HEPE is a bioactive lipid and has been implicated in various physiological processes, such as inflammation modulation and metabolic regulation [116]. Studies have observed that 12-HEPE can improve glucose metabolism by promoting glucose uptake, and decreased levels of 12-HEPE in obese patients suggest a potential link between 12-HEPE and metabolic health. Additionally, research in mice has shown consistent increases in 12-HEPE levels following cold exposure, which is known to activate brown fat [79].

#### 4.2.7. Alkylglycerols

Alkylglycerols (AKG), also known as alkylglycerol ethers, are a type of lipid molecule that is characterized by a glycerol backbone with one or more alkyl chains attached. Alkylglycerols are naturally found in certain biological tissues, such as the bone marrow, liver, and breast milk [117]. It induces beige fat development and protects mouse newborns from obesity [47,118]. Alkylglycerols are broken down by certain cells called adipose tissue macrophages (ATMs). This process leads to the release of interleukin-6 (IL-6) near adipocytes, prompting them to transform into a type of beige adipocyte. This communication between ATMs and adipocytes is governed by a signaling pathway called JAK/STAT3. This AKG-related signaling plays a crucial role during this sensitive phase of adipose tissue development and ultimately influences the likelihood of obesity in later life [47,119].

#### 4.2.8. Carnitine

Carnitine is a naturally occurring compound that plays a crucial role in energy metabolism within the cells of our body. It is involved in the transportation of long-chain fatty acids into the mitochondria, where they are oxidized to produce energy [120]. It is synthesized within the body from two essential amino acids, methionine, and lysine, in instances when it is not obtained through dietary sources. This synthesis can take place in various organs, such as the brain, liver, and kidneys [121]. In a study conducted on a mutant strain of mice with systemic carnitine deficiency, a reduction in lipid droplet formation and decreased energy metabolism were observed. However, these effects were found to be recoverable with l-carnitine supplementation [80]. Another finding observed in newborn goats indicates that maternal l-carnitine supplementation enhances BAT development and thermogenesis [81]. This underscores the significance of maternal nutrition in the development of mammalian infants. Despite the growing number of studies investigating the role of carnitine in BAT development, there remains a substantial need for further research. This includes exploring the underlying mechanisms of l-carnitine’s effects and potential gender differences.

## 5. Natural Products

### 5.1. Epigallocatechin Gallate (EGCG)

EGCG, a type of catechin found in green tea, has been shown to stimulate BAT thermogenesis and increase energy expenditure. A study found that EGCG supplementation in mice led to increased UCP1 expression and enhanced thermogenic capacity in brown adipocytes [82].

### 5.2. Resveratrol

Resveratrol is a polyphenol found in grapes, berries, and red wine. Research suggests that resveratrol may activate metabolic activity in brown adipocytes. Resveratrol administration to obese men resulted in activated AMPK increased SIRT1 and PGC-1α protein levels, which are key regulators of thermogenesis. Furthermore, resveratrol elevated intramyocellular lipid levels and decreased intrahepatic lipid content, circulating glucose, triglycerides, alanine–aminotransferase, and inflammation markers [83,122].

### 5.3. Caffeine

It is well established that caffeine is associated with weight loss, and in recent years, there has been increasing investigation into its potential influence on the activation of BAT. In vitro and in vivo studies discovered that caffeine increased the expression of thermogenesis-related genes such as PGC-1α and mitochondrial biogenesis, as well as levels of UCP1 in brown adipocytes [84]. Another study involving lean and obese individuals demonstrated that caffeine led to increased energy expenditure and lipid oxidation [85]. Additionally, a focused study observed that caffeine treatment resulted in increased expression of UCP1 and thermogenic genes in BAT, indicating the potential for BAT activation [86]. Another study also showed that higher maternal caffeine intake during pregnancy was associated with a higher risk of overweight/obesity and shorter stature in childhood. Specifically, children whose mothers had higher caffeine intake had higher levels of abdominal subcutaneous fat, visceral fat, and liver fat [123,124]. However, further research is needed to better understand the underlying mechanisms and to develop strategies for prevention and intervention.

### 5.4. Capsaicin and Capsinoids

Capsinoid is mainly found in red pepper. When ingested, it can increase energy expenditure in individuals with active BAT [125,126]. BAT-active individuals showed a stronger response to capsinoids compared to BAT-inactive individuals. This suggests that BAT is involved in capsinoid-induced energy expenditure, similar to findings in rodents. Capsaicin and capsinoids activate the TRPV1 receptor, which is known for perceiving capsaicin. Studies in mice have shown that capsaicin and capsinoids activate BAT thermogenesis and increase energy utilization through TRPV1 activation in the gastrointestinal tract. It is likely that orally ingested capsinoids activate BAT via gastrointestinal TRPV1 in humans [87]. A new discovery has come to light, indicating that TRPV1^+^ progenitor cells in BAT undergo thermogenic differentiation upon cold exposure [127]. This finding highlights the importance of exploring the potential role of capsaicin in promoting thermogenic differentiation in BAT.

## 6. Conclusions and Perspectives

This review provides an updated overview of information about thermogenic fat, such as BAT and beige fat, and how nutritional strategies activate their function and underlying mechanisms (Figure 2). Since the discovery of thermogenic fat, numerous advancements have been made in understanding how to activate brown/beige fat in human bodies to take advantage of its positive effects [13].

In recent years, there has been growing interest in the potential therapeutic applications of BAT and its activation for the treatment of metabolic disorders. Various approaches have been explored, including the use of drugs, dietary modifications, and transplantation [4,128,129]. While cold exposure is the most potent way to activate BAT/beige fat, it is not practical in tropical countries due to inconvenience. One drug that has shown promise in activating BAT is mirabegron, a β3-adrenergic receptor agonist originally approved for treating overactive bladder. Mirabegron has been found to increase BAT glucose uptake and energy expenditure in humans. However, high doses of mirabegron can lead to elevated blood pressure, limiting its clinical use. Dietary modifications, particularly the manipulation of macronutrients and micronutrients, have also been investigated for their potential to influence brown fat activity with fewer side effects. While the field of research on brown fat and its therapeutic implications is expanding rapidly, there are still many challenges to overcome. Identifying safe and effective methods for activating brown fat in humans, understanding potential side effects, and translating preclinical findings into effective clinical therapies are among the key areas that require further investigation.

Nutrition diet or strategy can play a crucial role in promoting the development and function of brown and beige fat, which may help address nutritional imbalances associated with their dysfunction. For example, a diet rich in PUFA has been shown to promote the development of brown and beige fat and improve metabolic health in animal models [129]. Additionally, certain dietary components, such as capsaicin found in chili peppers, can activate brown and beige fat thermogenesis and improve metabolic health [87]. Other dietary interventions, such as intermittent fasting and caloric restriction, have also been shown to promote the development and function of brown and beige fat. In summary, nutrition can be a potential solution for addressing nutritional imbalances associated with brown and beige fat dysfunction. A diet rich in PUFAs, capsaicin, and other dietary interventions such as intermittent fasting and caloric restriction can promote the development and function of brown and beige fat, which may improve metabolic health.

Proper nutrition plays a vital role in preserving well-being and safeguarding against illnesses. Although many animal studies have demonstrated the benefits, only a few nutritional diets or strategies have undergone clinical trials (Table 1), which may be due to challenges in balancing nutrients, determining appropriate dosages for individuals, and ensuring compliance with dietary restrictions. Despite these obstacles, conducting clinical trials remains important for evaluating the efficacy and safety of different diets and advancing evidence-based nutritional interventions.

Researchers are currently exploring various strategies to activate and recruit beige adipocytes in humans, such as cold exposure, exercise, pharmacological agents, and dietary interventions. Additionally, while ^18^F-fluorodeoxyglucose (^18^F-FDG) PET/CT is sufficient to detect brown adipocytes in the human body, a much-needed efficient diagnostic tool is still needed. In addition, much-needed research is still needed to fully understand the mechanisms behind beige adipocyte recruitment and its potential therapeutic applications. It is important to note that many of these studies provide preliminary evidence, and further research is needed to fully comprehend the mechanisms involved, but these findings have opened exciting new avenues for combating obesity and related diseases.

## Figures and Tables

**Figure 1 nutrients-15-04072-f001:**
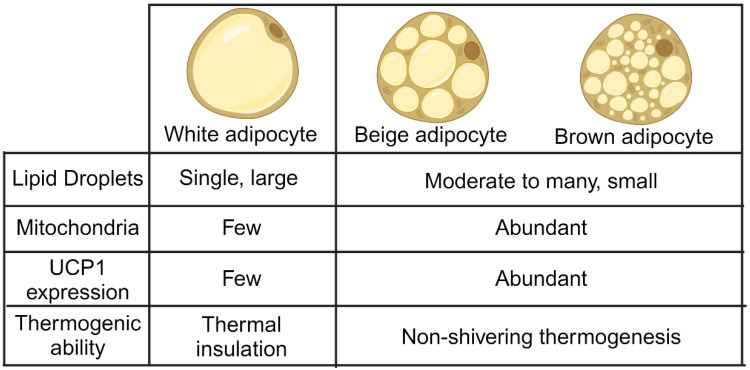
Differences between white, beige, and brown adipocytes. White adipocyte has large lipid droplets than beige/brown adipocytes, while the number of mitochondria is moderate to high in beige/brown adipocytes. Additionally, UCP1 expression is much lower in white adipocytes compared to beige/brown adipocytes. White adipocyte provides thermal insulation, and beige/brown adipocytes exert thermogenesis to maintain the body temperature.

**Figure 2 nutrients-15-04072-f002:**
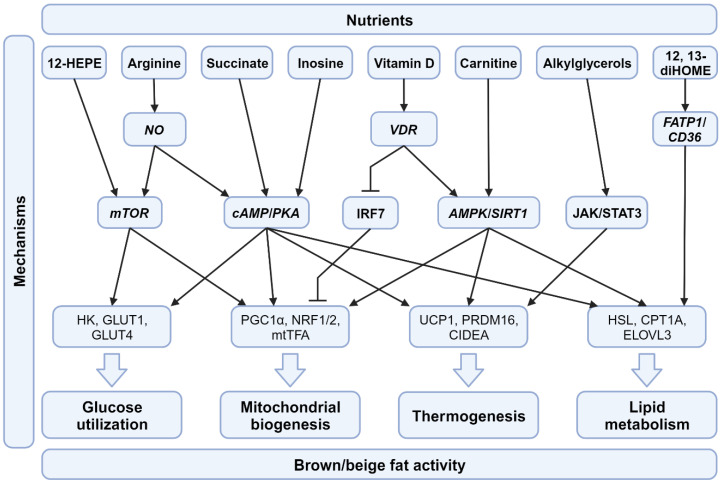
Molecular pathways involved in dietary supplements for BAT activation. Differential pathways initiated by various nutrients affect the functions of BAT, including glucose utilization, mitochondrial biogenesis, thermogenesis, and lipid metabolism. Abbreviations: NO, nitric oxide; VDR, vitamin D receptor; FATP1, fatty acid transport protein 1; CD36, cluster of differentiation 36; mTOR, mammalian target of rapamycin; cAMP, cyclic adenosine monophosphate; PKA, protein kinase A; JAK/STAT3, Janus kinase/signal transducer and activator of transcription 3; IRF7, IFN regulatory factor 7; AMPK, AMP-activated protein kinase; SIRT1, Sirtuin 1; HK, hexokinase; GLUT1, glucose transporter 1; GLUT4, glucose transporter 4; PGC1α, peroxisome proliferator-activated receptor gamma coactivator 1-alpha; NRF1/2, nuclear respiratory factor 1/2; mtTFA, mitochondrial transcription factor A; UCP1, uncoupling Protein 1; PRDM16, PR domain containing 16; CIDEA, cell death inducing DFFA like effector A; HSL; hormone-sensitive lipase; CTP1A, Carnitine palmitoyl transferase 1 A; ELOVL3, elongation of very long chain fatty acids protein 3.

**Table 1 nutrients-15-04072-t001:** Role of macronutrients, micronutrients, and natural products in BAT activity.

Types	Models	Impacts/Effects	Refs
**Macronutrients**			
Proteins	healthy men	↑ BAT activity	[69,70,71]
Carbohydrates	male mice	↑ thermogenesis and blood flow	[40]
Intermittent fasting	mice	↓ body weight; ↑ beige fat activity	[63]
Caloric restriction	mice	↓ body weight, leptin, glucose; ↑ beige fat and BAT activity	[66]
Omega-3 fatty acids	healthy men	↓ body weight; ↑ BAT activity	[72]
Omega-6 fatty acids	healthy men	↑ body weight; ↓ BAT activity	[73]
**Micronutrients**			
Vitamin D	obese rats	↓ body weight; ↑ WAT browning	[74]
Succinate	male mice	↓ body weight; ↑ glucose tolerance	[75]
Inosine	male mice	↓ body weight; ↑ glucose tolerance	[76]
Arginine	obese rats	↓ body fat; ↑ skeletal muscle and BAT mass	[77]
12,13-diHome	healthy men and women	↓ BMI and triglycerides; ↑ BAT activity	[78]
12-HEPE	male mice	↑ glucose tolerance	[79]
Alkylglycerols	infant mice	↓ fat accumulation; ↑ beige fat activity	[47]
Carnitine	juvenile visceral steatosis mice	↑ thermogenesis and BAT activity	[80]
	newborn goats	↑ thermogenesis and BAT activity	[81]
**Natural Product**			
Epigallocatechin gallate (EGCG)	male mice	↑ insulin sensitivity, thermogenesis, UCP1 level; ↓ blood glucose	[82]
Resveratrol	obese men	↓ triglycerides, glucose, energy expenditure	[83]
Caffeine	healthy men and women	↑ thermogenesis	[84]
	obese men	↑ energy expenditure and lipid oxidation	[85]
	obese mice	↓ body weight; ↑ glucose tolerance, UCP1 level	[86]
Capsaicin	healthy men	↑ Energy Expenditure, BAT activity	[87]

## Data Availability

No new data were created or analyzed in this study. Data sharing is not applicable to this article.

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
