# Peer review of "Brown Fat and Nutrition: Implications for Nutritional Interventions"

_nutrients, 2023, doi:10.3390/nu15184072_

Round 1
Reviewer 1 Report
I reviewed adipose tissue last year (for academic purposes).
The literature describes six phenotypes of adipose tissue; here, the authors report three in their overview.
In the literature, the beige adipose tissue is indicated as BAT-like. Authors here use the acronym BeAT, which is not usual in literature.
On pubmed searching for "beat" and "body composition" there are no works in which the beige adipose tissue is indicated with beats, therefore the authors cannot invent acronyms when they already exist of accepted (BAT-like).
Line 41/45: I ask the authors to verify better how many and which anti-obesity drugs are currently approved.
I think that authors should look at literature better.
Line 664: missing reference 111.
Reviewer 2 Report
The manuscript by Noriega et al., covers a timely and important topic. The concept of the manuscript is clear, and the structure is well thought and balanced. However, some shortcomings should be addressed before further consideration.
11. Authors overemphasize the role of cold exposure in the induction of thermogenic (brown or beige) fat. Please correct the text and Figure 1, as detailed here. In animals living under constant low temperature the UCP1 expressing thermogenic fat is lost, as heat insulation by white adipose tissue is a better adaptation strategy to cold than non-shivering thermogenesis [1] or, alternatively, non-shivering heat generation occurs in the skeletal muscle without the need for thermogenic fat [2]. Same applies to human population living in cold environment [3]. Thermogenic adipocytes are, hence, not the sole contributors to non-shivering thermogenesis, and the expansion of white adipocyte depots preserves core body temperature by offering an effective thermal insulation layer. Moreover, the human newborn has extensive thermogenic fat depots that do not require cold exposure to be functional [4, 5]. Similarly, beige adipocytes may develop without a cold stress, and are prevalent in some subcutaneous fat depots in the young [4, 6] and in the adult mouse [7]. Please correct this aspect in the text. Add a section on the thermogenic fat development in infancy. Key references are listed at the end of my comments. It has recently been suggested that adult humans have beige rather than brown adipocytes [8, 9].
22. Extend the section on the existence of thermogenic fat in children. The subcutaneous adipose tissue of newborns express UCP1 and other gene products associated with brown adipocytes [4-6, 10-13]. Similarly, the subcutaneous adipose tissue of the newborn mouse expresses gene products of mitobiogenesis and thermogenic fat differentiation [6]. In human infants, the level of UCP1 declines after 6 months of age and is sensitive to the type and duration of breastfeeding [4, 5].
33. One of the best characterized lipid mediators that induce thermogenic fat differentiation are alkylglycerols (AKGs) [4]. They induce thermogenic fat development in the infant, and they reduce obesity in adults by increasing thermogenesis and energy expenditure [4]. The molecular mechanism how AKGs induce thermogenic fat development is well known, however AKGs are not mentioned in the text. As they are easily available as food supplements, I believe this information is relevant for this review and it is key to address AKGs properly. Note that breastmilk is a key inducer of thermogenic fat development [4, 14], hence infant nutrition is a key mechanism that determines thermogenic capacity.
45. Mechanism of action of many of the listed metabolites are still undefined. Please specify this and provide some discussion on future directions of research.
56. Vitamin D has more complex and controversial role in adipose tissue. Please extend this section. It is known that obese individuals experience vitamin D deficiency [15, 16], however VDR is proposed to inhibit weight gain by activating UCP3 in the muscles [17]. Vitamin D synthesis is sunlight dependent, and a correlation exists between adult BMI and season of birth, and some studies have reported an increased rate of rate among winter-born individuals [18, 19]. And lastly, vitamin D / VDR signaling is key to promote mitochondrial biogenesis in thermogenic adipocytes by suppressing inflammation-inducing effect of mitochondrial contents [20]. Lack of VDR signaling disrupts this mechanism and leads to the loss of thermogenic capacity of adipocytes, causing obesity.
67. I suggest to remove the section about iron. The role of iron in thermogenic fat development is not clear. Consider discussing other – more relevant - metabolites, such as carnitine.
8. Please complete the section of 12,13-diHOME. It is present in human milk and determines adiposity of the infant. Also, correct the typo in the section title.
9. The text requires a careful editing. Redundancies and vague expressions are prevalent throughout the text. Examples: “Brown fat, known for its thermogenic properties and ability to produce heat through thermogenesis“ has redundant statements i.e. stating the heat generating fat generates heat through heat generation. There are many like this in the text.
Key references for the revision:
1. Gaudry, M.J., et al., Inactivation of thermogenic UCP1 as a historical contingency in multiple placental mammal clades. Science Advances, 2017. 3(7): p. e1602878.
2. Wright, T., et al., Skeletal muscle thermogenesis enables aquatic life in the smallest marine mammal. Science, 2021. 373(6551): p. 223-225.
3. Young, T.K., Obesity, Central Fat Patterning, and Their Metabolic Correlates among the Inuit of the Central Canadian Arctic. Human Biology, 1996. 68(2): p. 245-263.
4. Yu, H., et al., Breast milk alkylglycerols sustain beige adipocytes through adipose tissue macrophages. The Journal of Clinical Investigation, 2019. 129(6): p. 2485-2499.
5. Rockstroh, D., et al., Direct evidence of brown adipocytes in different fat depots in children. PLOS ONE, 2015. 10(2): p. e0117841.
6. Hoang, A.C., H. Yu, and T. Röszer, Transcriptional Landscaping Identifies a Beige Adipocyte Depot in the Newborn Mouse. Cells, 2021. 10(9): p. 2368.
7. Zhang, F., et al., An Adipose Tissue Atlas: An Image-Guided Identification of Human-like BAT and Beige Depots in Rodents. Cell Metab, 2018. 27(1): p. 252-262.e3.
8. Cannon, B., et al., Human brown adipose tissue: Classical brown rather than brite/beige? Experimental Physiology, 2020. 105(8): p. 1191-1200.
9. Sharp, L.Z., et al., Human BAT possesses molecular signatures that resemble beige/brite cells. PLoS One, 2012. 7(11): p. e49452.
10. Hahn, P. and M. Novak, Development of brown and white adipose tissue. Journal of Lipid Research, 1975. 16(2): p. 79-91.
11. Novak, M. and E. Monkus, Metabolism of Subcutaneous Adipose Tissue in the Immediate Postnatal Period of Human Newborns. 1. Developmental Changes in Lipolysis and Glycogen Content. Pediatric Research, 1972. 6(2): p. 73-80.
12. Novak, M., D. Penn, and E. Monkus, Regulation of lipolysis in human neonatal adipose tissue. Effects of alteration in carbohydrate metabolism. Biol Neonate, 1973. 22(5): p. 451-67.
13. Novak, M., E. Monkus, and V. Pardo, Human neonatal subcutaneous adipose tissue. Function and ultrastructure. Biol Neonate, 1971. 19(4): p. 306-21.
14. Tsukada, A., et al., White adipose tissue undergoes browning during preweaning period in association with microbiota formation in mice. iScience, 2023. 26(7).
15. de Oliveira, L.F., et al., Obesity and overweight decreases the effect of vitamin D supplementation in adults: systematic review and meta-analysis of randomized controlled trials. 2020. 21(1): p. 67-76.
16. Pramono, A., J.W.E. Jocken, and E.E. Blaak, Vitamin D deficiency in the aetiology of obesity-related insulin resistance. 2019. 35(5): p. e3146.
17. Fan, Y., et al., Vitamin D3/VDR resists diet-induced obesity by modulating UCP3 expression in muscles. Journal of biomedical science, 2016. 23(1): p. 56-56.
18. Wattie, N., C.I. Ardern, and J. Baker, Season of birth and prevalence of overweight and obesity in Canada. Early Hum Dev, 2008. 84(8): p. 539-47.
19. Phillips, D.I. and J.B. Young, Birth weight, climate at birth and the risk of obesity in adult life. Int J Obes Relat Metab Disord, 2000. 24(3): p. 281-7.
20. Hoang, A.C., et al., Mitochondrial RNA stimulates beige adipocyte development in young mice. Nature Metabolism, 2022.
Needs some editing,
Round 2
Reviewer 1 Report
Congratulations, you have clarified the doubts arising from the first reading of the paper. The current version is more understandable.
Line 44: High-
In the literature the beige adipose tissue is called: Beige or Brite or BAT-like. Term “brite” derives from “brown-in-white".
Frigolet ME, Gutiérrez-Aguilar R. The colors of adipose tissue. Gac Med Mex. 2020;156(2):142-149. English. doi: 10.24875/GMM.M20000356. PMID: 32285854.
Rabiee A. Beige Fat Maintenance; Toward a Sustained Metabolic Health. Front Endocrinol (Lausanne). 2020 Sep 4;11:634. doi: 10.3389/fendo.2020.00634. PMID: 33013707; PMCID: PMC7499124.
Reviewer 2 Report
The text has improved, only a few shortcomings exist still.
Abstract: “Brown fat, known for its thermogenic characteristic and ability to produce heat through 10 process called thermogenesis, can be activated through cold exposure. “ It is still a circular argument. „However, the practicality of this method is often limited.” This sentence has no meaning.
Please rephrase these lines. For example: „Beige and brown adipocytes produce heat in the adipose tissue, in a process that can be stimulated by cold exposure, hormonal signals and nutritional factors. Activation of thermogenic adipocytes holds promise in enhancing glucose metabolism, reducing fat accumulation, and improving insulin sensitivity. “
Lines 44-45: check grammar
Lines 88-89: Thermogenic adipocytes also store glycogen, especially after birth.
Lines 277-278: check grammar
Section on vitamin D: vitamin D supports mitochondrial biogenesis in beige fat of the newborn, that is a prerequisite of thermogenesis (Nature Metabolism, 4, 1684–1696 (2022)).
Line 375: AKGs affect only beige fat in mice (BAT in mice metabolizes AKGs into fatty acids)
Line 378: „…release of interleukin-6 (IL-6)…“
Section caffeine: please add a few lines on the risks of maternal coffee consumption on infant health
Line 469-473: language editing is needed
Figure 2: VDR blocks IFN-I signaling in beige adipocytes
Editing would be needed.
